# Corrosion Inhibition of Mild Steel in Hydrochloric Acid Environment Using Terephthaldehyde Based on Schiff Base: Gravimetric, Thermodynamic, and Computational Studies

**DOI:** 10.3390/molecules27154857

**Published:** 2022-07-29

**Authors:** Bahaa Sami Mahdi, Muna Khethier Abbass, Mustafa Khudhair Mohsin, Waleed Khalid Al-azzawi, Mahdi M. Hanoon, Mohammed Hliyil Hafiz Al-kaabi, Lina M. Shaker, Ahmed A. Al-amiery, Wan Nor Roslam Wan Isahak, Abdul Amir H. Kadhum, Mohd S. Takriff

**Affiliations:** 1Production Engineering and Metallurgy, University of Technology-Iraq, Baghdad 10001, Iraq; 70078@uotechnology.edu.iq (B.S.M.); muna.k.abbass@uotechnology.edu.iq (M.K.A.); 70182@uotechnology.edu.iq (M.M.H.); 2The Supreme National Authority for Accountability and Justice, Baghdad 10001, Iraq; faydalaa1991@gmail.com; 3Department of Medical Instruments Engineering Techniques, Al-Farahidi University, Baghdad 10001, Iraq; waleed.khalid@alfarahidiuc.edu.iq; 4College of Industrial Management of Oil and Gas, Basrah University of Oil and Gas, Basrah 61001, Iraq; drmhh1962@gmail.com; 5Department of Chemical and Process Engineering, Faculty of Engineering and Built Environment, Universiti Kebangsaan Malaysia (UKM), Bangi P.O. Box 43000, Malaysia; linamohmmed91@gmail.com; 6Energy and Renewable Energies Technology Center, University of Technology-Iraq, Baghdad 10001, Iraq; 7Faculty of Medicine, University of Al-Ameed, Karbala 56001, Iraq; amir1719@gmail.com; 8Chemical and Water Desalination Engineering Program, Department of Mechanical & Nuclear Engineering, College of Engineering, University of Sharjah, Sharjah 26666, United Arab Emirates; sobritakriff@ukm.edu.my

**Keywords:** terephthaldehyde, Schiff base, corrosion inhibitor, weight loss, DFT, EIS

## Abstract

Using traditional weight-loss tests, as well as different electrochemical techniques (potentiodynamic polarization and electrochemical impedance spectroscopy), we investigated the corrosion-inhibition performance of 2,2′-(1,4-phenylenebis(methanylylidene)) bis(N-(3-methoxyphenyl) hydrazinecarbothioamide) (PMBMH) as an inhibitor for mild steel in a 1 M hydrochloric acid solution. The maximum protection efficacy of 0.0005 M of PMBMH was 95%. Due to the creation of a protective adsorption layer instead of the adsorbed H_2_O molecules and acidic chloride ions, the existence of the investigated inhibitor reduced the corrosion rate and increased the inhibitory efficacy. The inhibition efficiency increased as the inhibitor concentration increased, but it decreased as the temperature increased. The PMBMH adsorption mode followed the Langmuir adsorption isotherm, with high adsorption-inhibition activity. Furthermore, the value of the $∆G_{ads}^{o}\;$ indicated that PMBMH contributed to the physical and chemical adsorption onto the mild-steel surface. Moreover, density functional theory (DFT) helped in the calculation of the quantum chemical parameters for finding the correlation between the inhibition activity and the molecular structure. The experimental and theoretical findings in this investigation are in good agreement.

## 1. Introduction

In view of its excellent mechanical durability, easy processing, and low production cost, iron and its alloys are applied as materials for construction in a variety of industrial uses, including petroleum, power plants, and chemical industries [1,2]. Steel comes into contact with corrosive environments in a variety of ways, including acidic solutions during etching, acid pickling, acid descaling, acid cleaning, and oil-well acidification [3]. Steel alloys react readily in acidic conditions, converting from a metallic to an ionic state, which results in significant economic loss. As a result, there is a pressing need to develop effective corrosion-control technologies. 

The use of corrosion inhibitors is one of the approaches [4,5,6]. Corrosion inhibitors are categorized based on their chemical structures, modes of action, and other factors. Organic corrosion inhibitors are among the most frequently used groups, and they have gained prominence due to their simplicity of synthesis at a low cost and their good protective capabilities. The preventative strategy is attributed to adsorption onto the mild-steel substrate, which obstructs corroding sites. The production of a protective barrier between the corrosive solution and the mild-steel surface prevents the mild steel from dissolving in the corrosive environment [7,8]. Corrosion is a significant issue in manufacturing, water, and circulation systems. Inhibitors are added to corrosive environments to prevent corrosion. 

Organic inhibitors include heteroatoms, such as N, O, S, and P, and they have been shown to be effective corrosion inhibitors in a variety of acidic solutions, both practically and theoretically [9,10]. The efficiency of these inhibitors can be related to their strong polarizability and decreased electronegativity, which allow them to cover vast metal surfaces and efficiently transfer electrons to the vacant orbitals of atoms [11]. Furthermore, organic inhibitors that contain nitrogen are effective corrosion-protection materials for alloys in HCl solutions, whereas sulfur-containing compounds are effective inhibitors in H_2_SO_4_ solutions. For both media, nitrogen- and sulfur-containing compounds act as ideal corrosion inhibitors [12]. The type of the distinctive inhibitor layer formed on the mild-steel substrate, as well as the quantity and nature of the adsorption centers attributed to the adsorption mechanism, determine the action of any inhibitor for any specific metallic alloy in severely corrosive solutions. 

Generally, the inhibitory efficiency of inhibitors with various heteroatoms follows the inverse sequence of the respective electronegativity values. Therefore, the inhibitory effectiveness of S, N, O, and P follows the order: oxygen (O) < nitrogen (N) < sulfur (S) < phosphorus (P) [13]. When organic molecules are present in acidic conditions, the electrochemical behavior of the acidic medium is frequently altered. In other words, it reduces the solution’s aggressiveness. Phosphorus, sulfur, oxygen, and nitrogen heteroatoms are found in the most commonly utilized heterocyclic compounds, and they effectively participate in the adsorption centers. We concentrated on the synthesis of a Schiff base generated from terephthaldehyde, namely, 2,2′-(1,4-phenylenebis(methanylylidene))bis(N-(3-methoxyphen yl)hydrazinecarbothioamide) (PMBMH), (Figure 1), as an acid corrosion inhibitor for mild steel, which works as a multifunctional protective agent. Terephthaldehyde and 4-(meta-methoxyphenyl)-3-thiosemicarbazide react to produce the chemical composition of the produced compounds. Because of these properties, the tested inhibitor has the potential to be a novel corrosion inhibitor. In continuation of the research on the advancement of Schiff bases as efficient anticorrosion compounds in HCl solution, the present investigation discusses the inhibition efficiency of PMBMH on mild-steel corrosion in a corrosive medium by using a weight-loss technique, potentiodynamic polarization, electrochemical impedance spectroscopy, and scanning electron microscopy. The experimental results are correlated with the quantum chemical calculations employing density functional theory (DFT) to explain how the inhibitor works to effectively coordinate with the mild-steel surface.

## 2. Results and Discussion

### 2.1. Gravimetric Techniques

The rates of the corrosion and inhibition efficiency of the tested corrosion inhibitor at different concentrations (0.0001, 0.0002, 0.0003, 0.0004, 0.0005, and 0.0010 M) and various temperatures (303–333 K) were analyzed by weight-loss measurements, and the experimental findings are presented in Figure 2 and Figure 3. PMBMH, the synthesized inhibitor, has been proven to be quite effective at inhibiting mild-steel corrosion. Figure 2 shows the effect of changing the PMBMH concentration on the corrosion rate, and thus the inhibition efficiency. At 303 K, the weight-loss measurements were taken. The corrosion rate decreased as the inhibitor concentration increased, and the inhibition improved, as indicated in Figure 2. This shows that, when the concentration rises, inhibitor molecules are essentially adsorbed to a higher extent onto the metal substrate, which results in broader surface coverage. As a result, it is clear that the performance of the inhibition was concentration dependent. Hence, the metal’s contact with the acid solution was limited. The PMBMH showed the highest inhibition efficacy at 0.0005, and the inhibition effectiveness was found to be 95.7%. 

Adsorbed inhibitor molecules control and/or block reaction sites, protecting the mild-steel surface from the corrosive medium. The inhibitor possesses many ion pairs of electrons, such as those on sulfur and nitrogen atoms, and the pi-electrons, which are coordinately bound with iron atoms on the coupon surface, which may be able to prevent corrosion. In other words, the excellent inhibition efficiency of PMBMH may be due to the addition of electron-donating functional groups, such as the amine, thione, imine, and aromatic rings, which improve the ability of inhibitor molecules to transfer electrons to the unoccupied d-orbitals of iron, thereby slowing the corrosion process. It is very well recognized that the effectiveness of inhibitor molecules increases with its electron-donating-power capacity. Increasing the inhibitor concentration does not result in a noticeable change in the percentage inhibition, as shown in Figure 2. This could be owing to the surface’s saturation. As a result, the optimum concentration is determined as 0.0005 M. 

The present inhibitor’s PMBMH inhibitory performance in HCl can be compared to those of earlier inhibitor studies that use heterocyclic organic molecules as corrosion inhibitors to protect metals from corrosive solutions. As shown in Table 1, most of the tested heterocyclic organic compounds have significant inhibitory efficacy. PMBMH has the strongest inhibitory ability of the heterocyclic organic compounds reported in Table 1 [14,15,16,17,18,19,20,21,22,23,24,25,26,27,28,29,30,31], with a performance equivalent to those reported in [15,23,26,28]. As the concentration of PMBMH increases, the rate of corrosion decreases, and the inhibitive efficacy improves. This could be because the inhibitor’s adsorption coverage on metal substrates increases as the PMBMH concentration rises.

### 2.2. Effect of Time

With a concentration of 0.0005 M, a temperature of 303 K, and a 5 h immersion time, the inhibitor had a maximum protection efficacy of 95.7%. The weight loss and corrosion rate increased with the immersion duration for the tested specimens, as well as at all the examined dosages of the examined inhibitor [31], according to the experimental observations shown in Figure 2. The corrosion rate increased marginally after 24 h of exposure in the presence of the studied inhibitor. As a result, the extended immersion duration in the acidic medium might be related to the minor increase in corrosion. For different inhibitor concentrations, Figure 2 displays a graph of the rate of corrosion and the protection performance against the immersion time (hours). The inhibition efficiency increased sharply with the lengthening of time for all inhibitor concentrations, as shown in these results. Then, based on the concentration of the inhibitor, a decrease in the value was detected after 24 h. The rate of corrosion increased marginally after 24 h of immersion with the addition of the examined inhibitor. As a consequence, the extended exposure duration in the HCl medium may be related to the slight increase in corrosion.

For different inhibitor concentrations, Figure 2 depicts a graph of the corrosion rate versus the protection efficacy (percentage) over time (hours). The protection efficacy increased sharply with increasing time for all the inhibitor concentration levels, as shown in these results. Then, based on the concentration of the inhibitor, a decrease in the value was detected after 24 h. The improvement in the protective efficacy is attributed to the increased surface coverage due to the adsorption of the inhibitor molecules onto the surface of the mild steel. By inhibiting the active sites, the adsorption barrier protects the mild-metal surfaces from the acid medium.

### 2.3. Effect of Temperature

Temperature has a significant impact on the corrosion rate and inhibiting efficacy, and particularly in an acidic medium. The rate of corrosion increases dramatically with rising temperature. The corrosion rate and inhibitory efficacy of the examined inhibitor were evaluated at varying temperature ranges (from 303 to 333 K) to fully understand its protection performance (Figure 3). At 303 K, the examined inhibitor had the maximum inhibitory efficacy, which gradually decreased as the temperature was raised. At the highest temperature, the tested inhibitor performed poorly in terms of protection. This finding indicates that the increase in temperature did not enable physical interactions (physisorption), which resulted in a reduction in the protective efficacy.

Arrhenius Equation (1) was used to calculate the activation parameters of the corrosion process (Figure 4):(1)$$logC_{R}=logK-\left(E_{a}/2.303RT\right)$$

The fact that inhibitory processes have a higher activation energy than uninhibited ones implies that the dissolution of mild steel takes time. Furthermore, when the inhibitor concentration increased, so did the activation energy. This means that the inhibitor acts as a physical barrier to the corrosion process, which becomes stronger as the concentration becomes higher. The adsorption of inhibitor molecules onto the metal surface reduces as the temperature rises, and, as a consequence, the corrosion rates rise. According to the modified Arrhenius equation (Equation (2)), from the graph of *log C_R_/T* versus *1/T* for the tested metal that dissolves in the corrosion solution, it is possible to find the values of the enthalpy of activation (Δ*H_a_*) and entropy of activation (Δ*S_a_*):(2)$$log\left\{\frac{C_{R}}{T}\right\}=\left\{log\left[\frac{R}{Nh}\right]+\left[\frac{∆S_{a}}{2.303R}\right]\right\}-\left[\frac{∆H_{a}}{2.303RT}\right]$$
where *N* is the Avogadro’s number, and *h* is the Planck’s constant.

A plot of *log C_R_/T* versus *1/T* is shown in Figure 5. The activation-enthalpy (Δ*H_a_*) and activation-entropy (Δ*S_a_*) values obtained from the slope of Figure 5, and the intercept, are shown in Table 2.

The endothermic nature of dissolving mild steel is demonstrated by the positive $∆H_{a}\;$ values. The low rate of corrosion is primarily due to the kinetic properties of activation, as indicated by the rise in the $∆H_{a}$ with the rising inhibitor concentration. The values of the activation enthalpy without and with the addition (0.0005 M) of the examined inhibitor were $\left(60.43\;\mathrm{kJ}\cdot {\mathrm{mol}}^{-1}\right)$ and $\left(43.76\;\mathrm{kJ}\cdot {\mathrm{mol}}^{-1}\right)$, respectively (Table 2), as estimated from the slope of Figure 5. The values of the activation entropy without and with the addition of the examined inhibitor were $\left(57.58\;J\cdot {\mathrm{mol}}^{-1}{\;K}^{-1}\right)$ and $\left(140.46\;J\cdot {\mathrm{mol}}^{-1}{\;K}^{-1}\right)$, respectively, as estimated based on the intercept of Figure 5. The mild-steel-dissolving process is endothermic, according to the positive $∆H_{a}$ values without and with the presence of the tested inhibitor.

### 2.4. Adsorption Isotherm

The quasi-substitution mechanism, according to Equation (3), which occurs at the electrode/electrolyte interface between the inhibitor molecules and the water or chloride ion, controls the competitive adsorption process:(3)$$Inh_{sol}+n{\left(H_{2}O\;or\;Cl^{-}\right)}_{ads}↔Inh_{ads}+n{\left(H_{2}O\;or\;Cl^{-}\right)}_{sol}$$
where $Inh_{sol}$ is the inhibitor molecules in the aqueous solution, and $Inh_{ads}$ is the adsorbed inhibitor molecules on the mild-steel surface.

The adsorption process is linked to the barrier-shielding layer covering the electrode surface (θ). Temkin, Frumkin, Flory–Huggins, and Langmuir adsorption isotherm models are used to fit the gravimetric measurements using the relationship (Equation (4)) between the inhibitor concentration $\left(C_{inh}\right)\;$ and the surface coverage (θ). The more suited adsorption model was developed (the Langmuir adsorption isotherm (as seen in Figure 6)) considering the high correlation coefficient $\left(R^{2}\right)$, which depicts the inhibitor-adsorption mechanism excellently, as per [32,33]:(4)$$\frac{C}{\theta }=\frac{1}{K_{ads}}+C$$
where θ represents the surface coverage, $K_{ads}$ refers to the constant of the adsorption process, and C is the concentration of the tested inhibitor.

The free adsorption energy $(∆G_{ads})\;$ was computed based on Equation (5):(5)$$∆G_{ads}=\left(\frac{1}{55.5}\right)exp\left(\frac{K_{ads}}{RT}\right)$$
where the value 55.5 represents the molar concentration of water, T is the absolute temperature, and R is the universal gas constant.

The greater the value of the $K_{ads}$, the stronger the inhibitor’s adsorption strength, and the lower the value of the $∆G_{ads}$, the more spontaneous the adsorption process. The larger $∆G_{ads}$ value shows that the tested inhibitor has a chemical adsorption activity in which electrons are transferred from high-electron centers to the unoccupied 3d orbital of iron, which creates coordination bonds. The negative value of the adsorption free-energy charge decreases as the molecular weight of the tested inhibitor increases ($-38.21\;\mathrm{kJ}\cdot {\mathrm{mol}}^{-1}$), which corresponds to the gravimetric observations [34,35].

### 2.5. Potentiodynamic Polarization Measurements

Figure 7 displays the polarization curves for the tested coupon in 1.0 M corrosive media with different PMBMH concentrations at 303 K. Table 3 lists the data for the inhibitory performances, corrosion potential ($E_{corr}$), and corrosion current density ($i_{corr}$), in addition to the anodic ($\beta_{a}$) and cathodic ($\beta_{c}$) Tafel slopes. These numbers were generated using the Tafel fit technique offered by the Gamry Echem analyzer program, which uses a nonlinear chi-squared minimization to fit the data to the Stern–Geary equation. Equation (6) represents how the inhibition efficiency was estimated:(6)$$IE\left(\%\right)=\frac{i_{corr}-i_{corr\left(inh\right)}}{i_{corr}}\times 100$$
where $i_{corr}\;$ is the corrosion current density in the absence of PMBMH, and $i_{corr\left(inh\right)}$ is the corrosion current density in the presence of PMBMH.

When a compound’s $E_{corr}$ displacement exceeds 85 mV with respect to the $E_{corr}^{o}$, it can be categorized as either a cathodic-type or anodic-type inhibitor [36,37]. The substances can be regarded as a mixed-type inhibitor because PMBMH causes the biggest displacement of the $E_{corr}$. Mild steel’s anodic dissolution is slowed down and cathodic hydrogen evolution is postponed when PMBMH is added to the corrosive environment. Table 3 demonstrates that the values of the $i_{corr}$ dropped when PMBMH was present. This finding suggests that the inhibition efficacy increases with the increasing PMBMH concentration, and the rate of corrosion reduces with the addition of the corrosion. The Tafel constant ($\beta_{a}$, $\beta_{c}$) values in the presence of PMBMH were barely varied, which proves that the PMBMH was in control of both processes. This finding also suggests that the hydrogen evolution or disintegration of mild steel was not impacted by the adsorbed molecules [38].

### 2.6. Electrochemical Measurements

Table 4 lists the experimental findings from the EIS analyses of mild-steel corrosion at 303 K, both without and with the addition of an inhibitor. Figure 8 displays Nyquist plots of the mild-steel-resistance spectrum in 1.0 M HCl in the absence and presence a varying PMBMH concentrations at 303 K. The inclusion of PMBMH caused the overall resistance to noticeably increase. Therefore, as seen in Figure 8, the presence of PMBMH in the HCl environment considerably altered the resistance responses of the mild steel. This variation was caused by an increase in the substrate resistance, which was related to an increase in the inhibitor’s concentration. The Nyquist plots showed two loops in the impedance curves of mild steel in the presence and absence of PMBMH: one loop in the high-frequency range (HF), and one loop at an intermediate frequency (MF). Furthermore, at a lower frequency (LF), negligible inductive action was seen. The EIS instrument’s limitations at high frequencies with low resistances and charge-transfer operations were cited as the causes of the HF and MF loops, respectively. The relaxing mechanism of the adsorption of corrosion products or the adsorption of inhibitor molecules onto the mild-steel surface in acidic solution with and without the inhibitor, respectively, was said to be the cause of the inductive behavior seen in the LF region [39]. The following equation (Equation (7)) was used to obtain the inhibition efficiency (IE%) from the charge-transfer impedance:(7)$$IE\left(\%\right)=\frac{R_{ct}^{\prime }-R_{ct}}{R_{ct}^{\prime }}\times 100$$
where $R_{ct}^{\prime }$ is the charge-transfer-resistance value in the presence of PMBMH, and $R_{ct\;}$ is the charge-transfer-resistance value in the absence of PMBMH.

In the presence of PMBMH, it was discovered that the $C_{dl}$ value rises from 323 ${\mu F\;\mathrm{cm}}^{-2}$ (0.0004 M) to 475 ${\mu F\;\mathrm{cm}}^{-2}$ (0.0005 M), which suggests that the layer thickness at 0.0005 M is greater than that at 0.0004 M.

According to Table 4, the charge-transfer resistance ($R_{ct}$) improved along with the inhibitor concentration. Systems that corrode gradually are attributed to large charge-transfer impedances [40]. Additionally, reduced metal capacitance is linked to enhanced inhibitor resistance. The observed rise in the $C_{dl}$, which was attributed to an increase in the local dielectric constant and/or the electrical double layer’s thickness, indicates that the PMBMH adsorbed onto the interface between the material and the environment [41]. Inhibitor adsorption onto the most active adsorption sites may be to blame for the observed increase in the $C_{dl}\;$ value in a corrosive solution in the presence of increasing concentrations of PMBMH [42]. The homogeneity of the adsorbed PMBMH layer reduced due to the corrosion reaction.

Additionally, the findings demonstrated that, when the inhibitor concentration increased, so did the IE percent. This outcome exhibits the same pattern as the IE percent discovered through the weight-loss and potentiodynamic measurements. The equivalent circuit diagram for fitting the actual EIS results for hydrochloric acid in the presence and absence of the inhibitor is shown in Figure 9A,B. The solution resistance ($R_{s}$), a constant-phase element $\left(CPE_{dl}\right)$, and the charge-transfer resistance are the circuit components for the data (Rct). The Rct value provided an indication of how well the electrons were moving over the contact. Rad was utilized as a model for the inhibitor adsorption in conjunction with a $\left(CPE_{ad}\right)$ [43]:

In Figure 9, $R_{s}$ is the environment impedance; Rct is the charge-transfer impedance; $CPE_{dl}\;$ signifies the double-layer constant-phase element; $R_{ad}$ refers to the adsorbed-layer impedance; $CPE_{ad}\;$ is the adsorbed-layer constant-phase element. Because an equivalent circuit is used to determine the simulated values and for comparison with experimental data, a CNLS (complex nonlinear least squares) simulation was used, as previously reported [44,45,46].

### 2.7. Surface Study

The scanning-electron-microscope (SEM) pictures of the mild-steel-coupon surface after immersion in the 1 M HCl medium, having (A) no inhibitor and (B) 0.0005 M PMBMH, for 5 h at 303 K, are demonstrated in Figure 10. The mild-steel-coupon surface corroded severely in the corrosive solution, as shown by the rough surface of the coupon surface in Figure 10A. The abundance of corrosion porous products can be obviously viewed on the mild-steel-coupon surface in Figure 10A. On the one hand, the mild-steel-coupon surface was damaged when immersed in the HCl environment, as demonstrated by Figure 10A. On the other hand, Figure 10B’s depiction of the presence of an inhibitor shows that the mild-steel surface remained noticeably undamaged, inhibiting the corrosion process. This discovery confirmed the outcomes of the other experiments. As observed in Figure 10B, the mild-steel coupon immersed in 1 M HCl solution (in the presence of 0.0005 M of the tested inhibitor) had a smoother texture than the coupon exposed to the uninhibited acid solution, which indicates that the presence of the inhibitor protected the coupon surface from the corrosive environment’s attack.

### 2.8. Quantum Chemical Computations

With the use of ChemOffice software, the adsorption ability of the synthesized inhibitor over the mild-steel-coupon surface can be explored from a quantum-chemical-computation standpoint, using frontier molecular orbital theory (FMOT) [47,48,49]. The donor–acceptor interactions could be at the root of the tested inhibitor’s adsorption. In this situation, electrons are moved from the organic compound’s high-electron-density centers (nitrogen, oxygen, sulfur, and electrons (i.e., regions of highly electronic distributions)) to the mild-steel surface’s unoccupied 3d orbital (iron surface).

Table 5 depicts and discusses the quantum chemical variables acquired from the optimization of the tested inhibitor in the gas phase (shown in Figure 11). The highest occupied-molecular-orbital (E_HOMO_) energy shows the capability of the tested inhibitor to donate electrons. The capability of the molecules to accept electrons from the back donation of iron, and to thus enhance the binding energy between the metal and the inhibitor, is shown by a lower ELUMO value.

The greater the E_HOMO_ and the lower the E_LUMO_, the better the tested inhibitor’s capacity to attach to the metal surface [50,51,52,53]. As seen in Table 5, the E_HOMO_ rises as the number of examined inhibitor compounds rises. This agrees well with the experimental data and implies that PMBMH is an effective corrosion inhibitor [54]. The energy gap (E_gap_) values determine the chemical response. In terms of the reactivity, the greater the inhibiting efficiency of the molecule, the more reactive it is toward the substrate surface, and the smaller the Δ*E* gap, the more stable it is. As a result, PMBMH and the Fe substrate create a stable combination. The dipole moment (μ) is caused by the atoms in the molecule having a nonuniform-distribution surface charge. The value of the dipole moment is decreasing, which supports the inhibitor molecule. In the hard–soft acid–base notion, the soft molecule has lower E_gap_ values and higher basicity, and the opposite is true when comparing the hard ones [55]. As a result, the soft molecule has more adsorption ability due to its easier electron transfer, and it is a stronger corrosion inhibitor than the hard molecule.

According to Lukovit’s research, when the number of electrons transmitted (ΔN) is less than 3.6, the inhibition performance improves as a function of the electron transfer [56]. The greater the corrosion inhibitor, the larger the fraction of electron transport (ΔN). As shown in Table 5, the ΔN increases as the quantity of ethylene oxide units increases, which confirms the highest adsorption ability of the tested inhibitor, as previously stated by the experimental data. The chemical reactivity of atoms and molecules is represented by the ionization energy. As a result, as the ionization energy rises, the reactivity rises as well, and the inhibition efficiency falls. These data back up the gravimetric readings that were taken.

### 2.9. Mulliken Charges

Mulliken charges were used to determine the adsorption centers of the tested inhibitor [57]. The easier it is for the atoms to transfer electrons, the more negatively charged they are. Table 6 shows that N, O, and S atoms are electron-donating sites in the inhibitor compounds that have been studied. It is obvious from Table 6 that S(25), O(33), N(22), O(20), N(13), S(12), and N(9) have the highest atomic charges (−0.3009, −0.2084, −0.2717, −0.2126, −0.2595, −0.2816, and −0.2638, respectively). These atoms have the ability to bond with the metal surface and form coordination bonds.

### 2.10. Suggested Corrosion-Inhibition Mechanism

The creation of a protective layer that is absorbed onto the iron surface is responsible for the inhibitory inertia of organic molecules. The presence of the tested inhibitor significantly reduced the mild-steel corrosion, according to gravimetric tests. Furthermore, the adsorption isotherm analyses show that the adsorption of the studied inhibitor molecules onto the mild-steel surface follows the Langmuir adsorption model quite closely. Furthermore, the generated protective film’s adsorption behavior is principally determined by: (1) electrostatic interactions through protonated heteroatoms, and (2) different linkages between inhibitor molecules and the mild-steel surface [58].

The interaction between the tested inhibitor molecules and the mild-steel surface is depicted in further detail in Figure 12. The interaction between both the pi-electrons of the aromatic rings and the empty d-orbital of the metal atoms is the most common technique of adsorption between the inhibitor molecule and the mild-steel surface. The second method is the donor–acceptor interactions between the vacant d-orbital of the iron (mild steel) surface atoms and the lone electron pairs in the heteroatoms (S, O, and N). These active electrons are destined to be shared with the Fe atom’s d-orbitals.

## 3. Materials and Methods

### 3.1. Synthesis of the Corrosion Inhibitor

A solution of terephthaldehyde (0.1 mmol) in 100 mL ethyl alcohol was refluxed with 4-(meta-methoxyphenyl)-3-thiosemicarbazide (0.2 mmol) for 5 h with a few drops of glacial acetic acid. After cooling to room temperature, a solid mass was separated and recrystallized from ethyl alcohol with an 83% yield. The yellow product (221–223 °C) was characterized by various spectroscopic techniques. IR: 3289.5 and 3139.6 cm^−1^ (N–H, amines), 3079.3 cm^−1^ (C–H, aromatic), 2945.3 cm^−1^ (C–H, aliphatic), 1548.6 cm^−1^ (C=N, imine), 1463.8 cm^−1^ (C=C, aromatic); 1168.7 and 1051.5 cm^−1^ (C-O, sy and asy). ^1^H-NMR (DMSO-d^6^): δ 3.73 (6H, s, -OCH_3_), 6.65 (2H, dd), 7.17–7.46 (6H, d, aromatic ring), 7.57 (4H, dd, aromatic ring), 7.94 (2H, s, -CH=N-), 9.93 (2H, s, NH), and 11.47 (2H, s, NH). ^13^C NMR (DMSO-d^6^): δ 57.2, 107.1, 114.9, 121.3, 129.6, 131.2, 133.7, 134.1, 139.1, 139.9, 159.4, 178.3.

### 3.2. Mild Steel

The mild-steel-sample elemental analysis used for the gravimeter measurements (wt%) is detailed in Table 7.

### 3.3. Medium

The solubility of PMBMH was determined in 1 M hydrochloric acid solution. The 1 M corrosive environment was prepared by diluting 37% HCl (Merck-Malaysia) in double-distilled water. The measurements were performed without and with the addition of PMBMH at different concentrations (0.0001, 0.0002, 0.0003, 0.0004, 0.0005, and 0.001 M), with various immersion times (1, 5, 10, 24, and 48 h). The effect of temperature (303, 313, 323, and 333 K) was also studied at various inhibitor concentrations for an immersion time of 5 h.

### 3.4. Weight-Loss Measurements

The traditional weight-loss measurements were performed according to the procedures described in this article [59,60,61]. At temperatures ranging from 303 K to 333 K, cleaned, polished, and weighed mild-steel specimens were immersed in a 1 M HCl environment in the absence and presence of varied doses of the tested inhibitor for various exposure times (1, 5, 10, 24, and 48 h). To ensure that the tests could be replicated, each measurement was carried out three times, and only the average results were published. The corrosion rate and inhibition efficiency were calculated from Equations (8) and (9):(8)$$C_{R}=\frac{87600W}{atd}\;\;$$
where *W* is the weight loss of the tested mild steel (gram), a is the tested mild-steel surface area (cm^2^), *t* is the exposure time (h), and *d* is the tested mild-steel density (gcm^−3^);
(9)$$IE\%=\left[\frac{C_{R\left(o\right)}-C_{R\left(i\right)}}{C_{R\left(o\right)}}\times 100\right]\;$$
where $C_{R\left(o\right)}$ is the rate of corrosion in the absence of PMBMH, and $C_{R\left(i\right)}$ is the rate of corrosion in the presence of PMBMH.

### 3.5. Electrochemical Data

As the working electrodes throughout the study, mild-steel coupons were cleaned in accordance with ASTM standard G1-03 [62]. The coupon’s active surface area was 4.5 cm^2^. The experiments were performed in a 1.0 M HCl environment that was aerated but not agitated at 303 K, with tested inhibitor concentrations ranging from 0.0001 to 0.0005 M. To ensure that the tests could be replicated, each measurement was carried out three times, and only the average results were published. A Gamry Instrument Potentiostat/Galvanostat/ZRA type REF 600 was used for the measurements. Gamry’s DC105 and EIS300 software was utilized for the electrochemical impedance spectroscopy (EIS) and potentiodynamic scanning, respectively. At a scan rate of 0.5 mVs^−1^, the potentiodynamic current-potential curves were adjusted from 0.25 to +0.25 V sce. Using the Gamry Echem Analyst program, all of the impedance values were matched to the proper equivalent circuits (ECs). A Gamry water-jacketed glass cell with three electrodes—the working electrode, the counter electrode, and the reference electrode—was used to assess the inhibitory effects of the tested inhibitor. The reference electrode was a saturated calomel electrode (SCE). Electrochemical tests were started about 30 min after the working electrode was submerged in the solution in order to maintain the steady-state potential [44,63,64].

### 3.6. Surface Scanning Electron Microscope

Scanning electron microscopy was used to observe the corrosive behavior of the acidic solution (1 M HCl) on the mild-steel surface after 5 h of treatment without and with the addition of 0.0005 M of PMBMH (SEM, Zeiss MERLIN Compact FESEM at the UKM Electron Microscopy Unit).

### 3.7. Computational Studies

Conceptual computations have recently been performed to simulate experimental findings. ChemOffice was used to perform the quantum chemical computations. We used the conventional theory of Becke’s three-parameter hybrid functional (B3LYP) level using a Gaussian 03 version, with 6–31G as the reference set, to investigate the chemical reactivity of the PMBMH molecule. For the synthesized compound (PMBMH in the gas phase), the investigated approach produces the quantum parameters of the energy gap (ΔE), fraction of electron transfer (ΔN), dipole moment (μ), e ionization energy (I), electron affinity (A), absolute electronegativity (χ), ionization energy (I), hardness (η), and softness (σ) [65].

The quantum chemical parameters, such as *χ*, *η*, *σ*, and ΔN, were calculated from Equations (10)–(15):(10)$$I=-E_{HOMO}$$
(11)$$A=-E_{LUMO}$$
(12)$$\chi =\frac{I+A}{2}$$
(13)$$\eta =\frac{I-A}{2}$$
(14)$$\sigma =\frac{1}{\eta }$$
(15)$$∆N=\frac{\left(\chi_{Fe}-\chi_{inh}\right)}{2\left(\eta_{Fe}-\eta_{inh}\right)}$$

## 4. Conclusions

Due to the existence of very effective electronic-adsorption centers (S, O, N, and pi-bonds) that block the active centers of iron metal, the examined corrosion inhibitor shows a good corrosion-protection performance for mild steel in 1 M of HCl. The anticorrosion efficiency of PMBMH was studied by weight loss, potentiodynamic polarization, and electrochemical measurements, in addition to quantum chemical calculation. The molecular structure, concentration, and temperature parameters that effect the corrosion inhibition were examined. The following are the key findings:PMBMH has the maximum efficiency of 95% at 303 K and is a potential steel corrosion inhibitor in 1.0 M HCl. The heteroatoms and benzene rings in the PMBMH’s structure are responsible for its considerable inhibitory efficiency;Gravimetric tests showed that the inhibition performance increases with the concentration, with the tested corrosion inhibitor’s peak inhibition efficiency being 95% at the ideal concentration (0.0005 M);The inhibitor tends to be weakly bonded to the metal surface, and the efficacy of inhibition reduces as the temperature increases. On metallic surfaces, it is anticipated that it will take part in chemical adsorption;The electrochemical-experiment findings indicated that PMBMH shields mild-steel corrosion because a shielding inhibitor coating forms at the steel–electrolyte interface;The value of the G_ads_ obtained is used to propose chemisorption and physisorption phenomena, and the adsorption process is spontaneous;Quantum chemistry simulations demonstrate that the tested inhibitor molecules adsorb onto a mild-steel surface utilizing sulfur, oxygen, and nitrogen heteroatoms, in addition to the pi-electrons of the benzene rings, as their efficient centers;As a result, the findings of the experiments and theoretical analysis are in good accord.

## Figures and Tables

**Figure 1 molecules-27-04857-f001:**
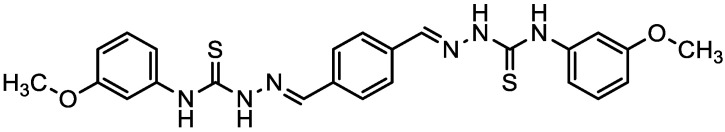
The chemical structure of PMBMH.

**Figure 2 molecules-27-04857-f002:**
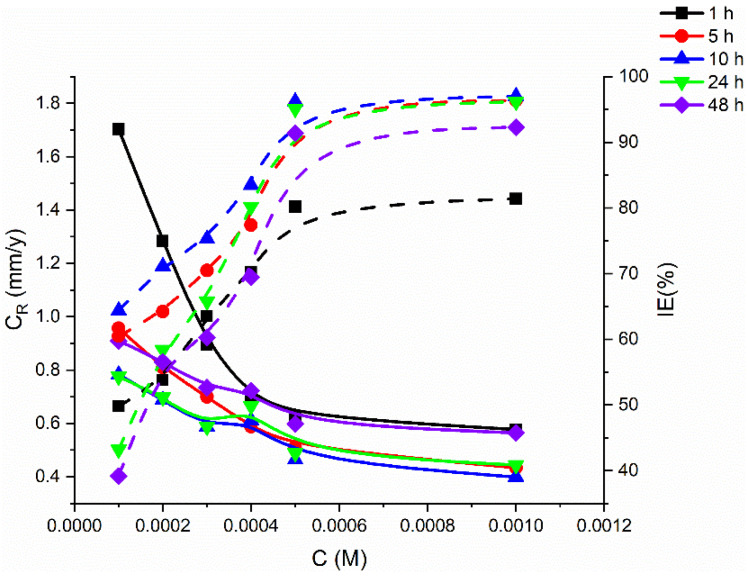
Gravimetric-curve relationship of metal coupons in 1 M HCl between corrosion rate and inhibition efficiency: different exposure periods at 303 K.

**Figure 3 molecules-27-04857-f003:**
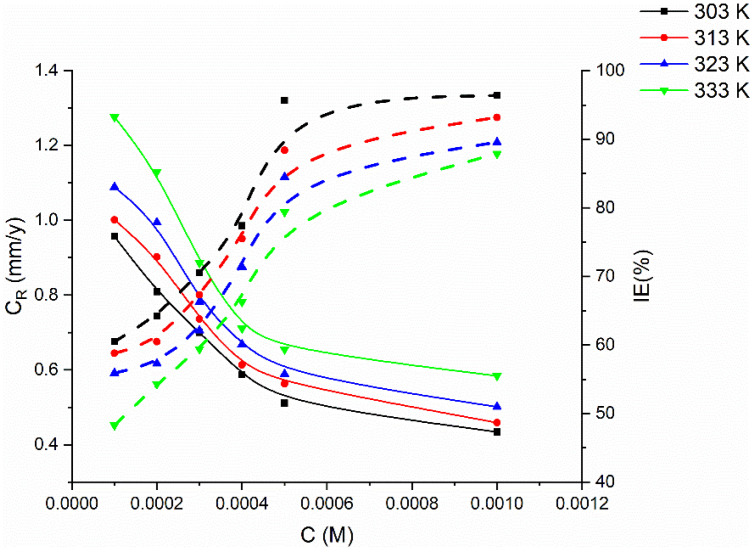
Gravimetric-curve relationship of metal coupons in 1 M HCl between corrosion rate and inhibition efficiency against concentration of PMBMH at different temperatures for 5 h exposure period.

**Figure 4 molecules-27-04857-f004:**
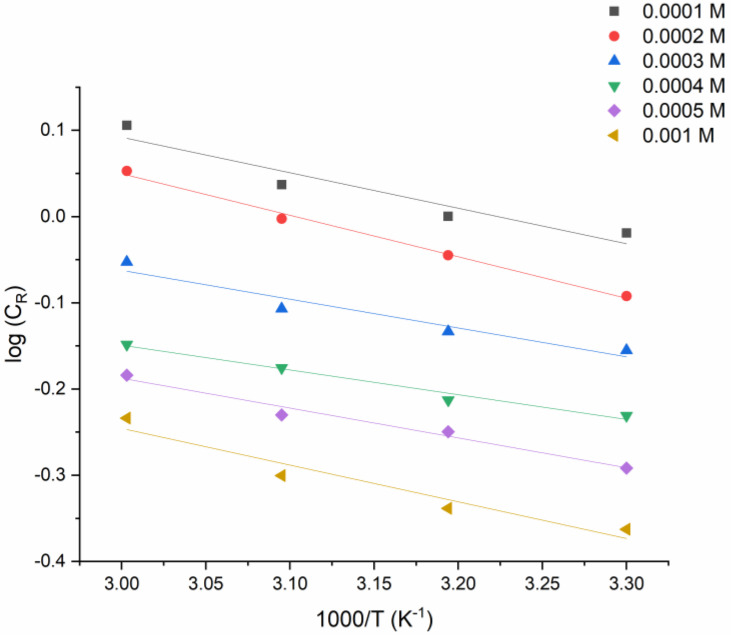
The log (C_R_) versus 1/T graph for the various concentrations of PMBMH and different temperatures.

**Figure 5 molecules-27-04857-f005:**
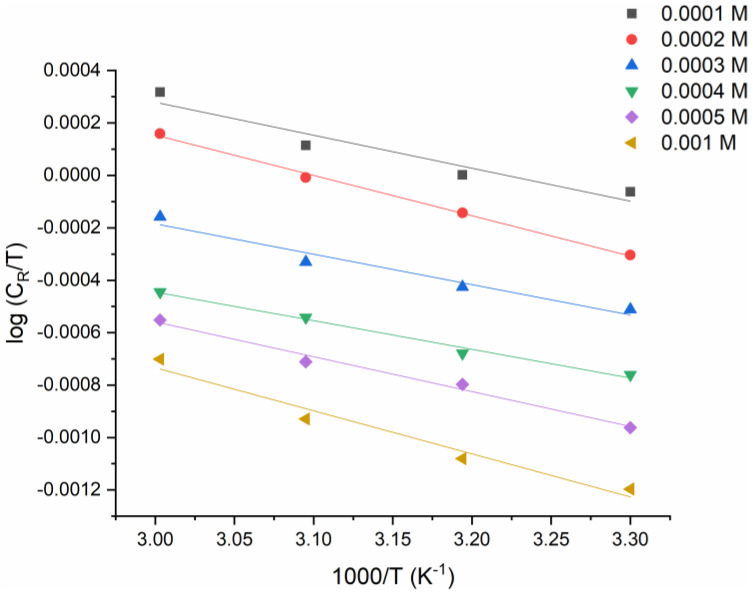
A plot of Arrhenius modified equations of $log\left\{\frac{C_{R}}{T}\right\}\;$ versus $\frac{i}{T}$ for tested metal with different concentrations of the examined inhibitor.

**Figure 6 molecules-27-04857-f006:**
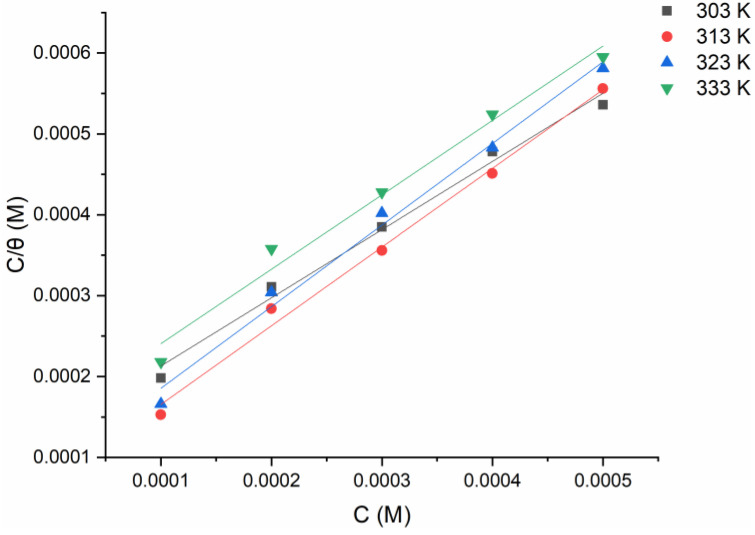
Langmuir adsorption model of tested inhibitor on the surface of mild steel in 1 M HCl at 303 K from the gravimetric data.

**Figure 7 molecules-27-04857-f007:**
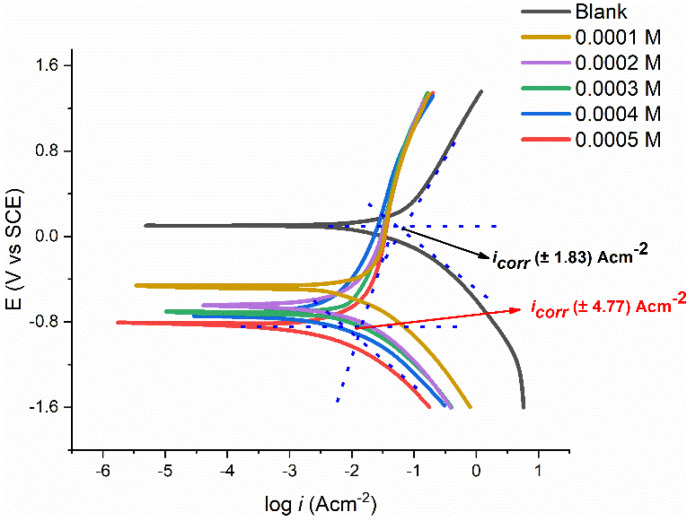
Polarization curves of tested coupons in 1 M HCl solution with different concentrations of PMBMH.

**Figure 8 molecules-27-04857-f008:**
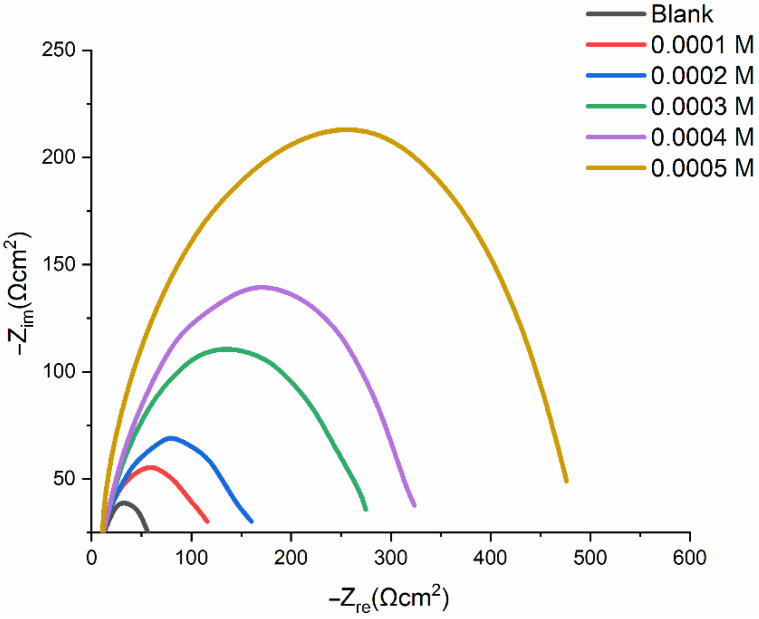
Nyquist plots of mild steel in 1 M HCl without and with the addition of various concentrations of PMBMH.

**Figure 9 molecules-27-04857-f009:**
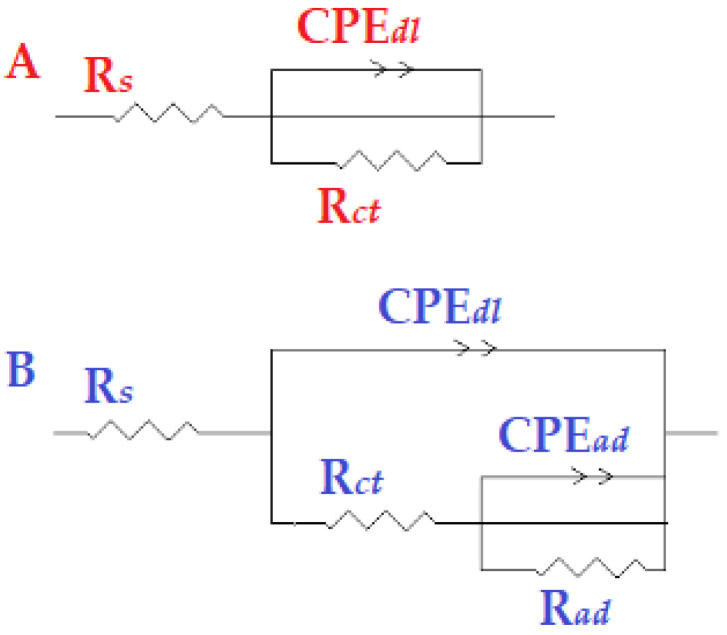
The model of equivalent circuit that was used to fit the experimental data (**A**) without and (**B**) with the addition of the tested inhibitor.

**Figure 10 molecules-27-04857-f010:**
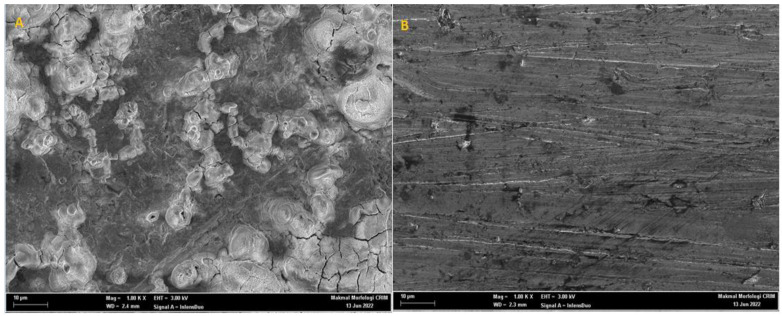
SEM micrographs showing the surface morphology of mild-steel-coupon surface in absence (**A**) and presence (**B**) of 0.0005 M PMBMH in 1 M HCl environment for 5 h at 303 K.

**Figure 11 molecules-27-04857-f011:**
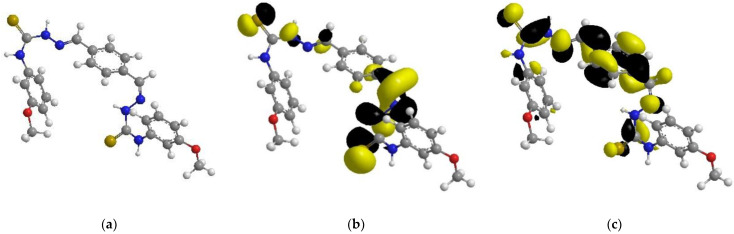
The optimized chemical structure (**a**), highest occupied molecular orbital (**b**), and lowest unoccupied molecular orbital (**c**) of the tested inhibitor.

**Figure 12 molecules-27-04857-f012:**
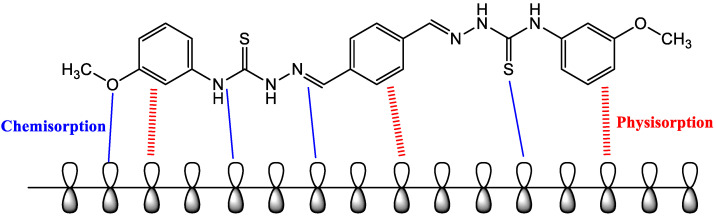
Suggested corrosion-inhibition mechanism of mild steel in 1 M HCl with the addition of the examined inhibitor.

**Table 1 molecules-27-04857-t001:** PMBMH inhibition was statistically compared to the efficacy of other heterocyclic organic inhibitors that had previously been investigated.

Corrosion Inhibitor	Metal	Acid	IE%	Ref.
PMBMH	Mild steel	HCl	95.7	-
(E)-N0 -(2,4-dimethoxybenzylidene)-2-(6-methoxynaphthalen-2-yl) propanehydrazide	Mild steel	HCl	95	[14]
N0-cyclohexylidene-2-(6-methoxynaphthalen-2-yl) propanehydrazide	Mild steel	HCl	86	[14]
Furoin thiosemicarbazone	Mild steel	HCl	98.7	[15]
2-pyridinecarboxaldehyde thiosemicarbazone	Mild steel	HCl	89.7	[16]
4-pyridinecarboxaldehyde thiosemicarbazone	Mild steel	HCl	85.5	[16]
clozapine	Mild steel	HCl	96	[17]
5-hexylsulfanyl-1,2,4-triazole	Mild steel	HCl	97	[18]
triglycidyl ether of triethoxy triazine	Carbon steel	HCl	88	[19]
N-propargyl saccharin	C38 steel	HCl	90	[20]
N-(2-aminophenyl)-2-(5-methyl-1H-pyrazol-3-yl) acetamide	C38 steel	HCl	93	[21]
dodecyl vanillin–glycine Schiff base	Carbon steel	H2SO4	74.4	[22]
hexadecyl vanillin–glycine Schiff base	Carbon steel	HCl	78.5	[22]
octadecyl vanillin–glycine Schiff base	Carbon steel	HCl	80.8	[22]
oleic acid vanillin–glycine Schiff base	Carbon steel	HCl	81.9	[22]
3-(3-formyl-4-hydroxy- l-phenylazo) - l, 2, 4-triazole	Copper	HNO3	97	[23]
3-(2-hydroxy-5-methyl- l -phenylazo) - l, 2, 4-triazole	Copper	HNO3	95.8	[23]
3-(4-hydroxy-l-phenylazo)-l, 2, 4-triazole	Copper	HNO3	97.2	[23]
2-amino-7-hydroxy-4-phenyl-1,4- dihydroquinoline-3-carbonitrile	Mild steel	HCl	93.3	[24]
2-amino-7-hydroxy-4-(p-tolyl)-1,4 dihydroquinoline-3-carbonitrile	Mild steel	HCl	92.8	[24]
2- , 2-amino-7-hydroxy-4-(4-methoxyphenyl)-1,4 dihydroquinoline-3 carbonitrile	Mild steel	HCl	96.6	[24]
2-amino-4-(4- (dimethylamino)phenyl)-7-hydroxy-1,4-dihydroquinoline-3-carbonitrile	Mild steel	HCl	98	[24]
y 5-Styryl-2,7- dithioxo-2,3,5,6,7,8- hexahydropyrimido [4,5-d] pyrimidin-4(1H) one	Carbon steel	HCl	89.1	[25]
5-(2- Hydroxyphenyl)-2,7-dithioxo-2,3,5,6,7,8- hexahydropyrimido [4,5-d]-pyrimidin-4(1H) one	N80 steel	HCl	73.1	[25]
5-(2,5-dimethylthiophen-3yl)-4-(4-(6-(2,5-dimethylthiophen-3-yl)-2-hydroxypyrimidin-4- yl)phenyl)pyrimidin-2-ol	Mild steel	H_2_SO_4_	98.3	[26]
5-(2,5-dimethylthiophen-3yl)-4-(4-(6-(2,5-dimethylthiophen-3-yl)-2-mercaptopyrimidin-4- yl)phenyl) pyrimidin-2-thiol	Mild steel	H_2_SO_4_	99.3	[26]
2-chloropyrimidine	Cold rolled steel	HNO_3_	14.5	[27]
2-hydroxypyrimidine	Cold rolled steel	HNO_3_	23.0	[27]
2-bromopyrimidine	Cold rolled steel	HNO_3_	27.2	[27]
2-aminopyrimidine	Cold rolled steel	HNO_3_	35.0	[27]
2-mercaptopyrimidine	Cold rolled steel	HNO_3_	99.1	[27]
2-((6-methyl-2-ketoquinoUne-3-yl)methylene) hydrazinecarbothioamide	Mild steel	HCl	95.8	[28]
4-(6-methylcoumarin)acetohydrazide	Mild steel	HCl	94.5	[29]
4-(Benzoimidazole-2-yl)pyridine	Mild steel	HCl	93.8	[30]
5,5′-(1,4-phenylene)bis(N-phenyl-1,3,4-thiadiazol-2-amine)	Mild steel	HCl	94	[31]

**Table 2 molecules-27-04857-t002:** Isothermal parameter values for mild-steel coupon in 1 M HCl in the absence and presence of different concentrations of PMBMH.

C (M)	$E_{a}\;\left(kJ\cdot mol^{-1}\right)$	$∆H_{a}\;\left(kJ\cdot mol^{-1}\right)$	$∆S_{a}\;\left(J\cdot mol^{-1}\;K^{-1}\right)$
Blank	63.11	60.43	57.58
0.0001	51.46	52.65	123.54
0.0002	47.85	50.45	120.43
0.0003	45.35	47.67	124.43
0.0004	43.63	45.73	131.84
0.0005	41.78	43.76	140.46
0.001	39.85	42.10	151.65

**Table 3 molecules-27-04857-t003:** Tafel parameters for tested coupon without and with the addition of various concentrations of PMBMH in 1 M HCl solution.

Conc. M	*E_corr_* (V)	β_a_ (mV/dec)	β_c_ (mV/dec)	* i *_ corr _ (μA·cm^−2^)	IE (%)
0.000	–0.46	120	140	663.8 ± 1.83	0
0.0001	–0.49	118.7	131.5	400.3 ± 5.03	67.1
0.0002	–0.52	131.5	133.2	230.1 ± 3.70	73.5
0.0003	–0.54	88.5	131.6	110.4 ± 2.93	84.7
0.0004	–0.51	67.5	100.5	90.4 ± 1.84	87.4
0.0005	–0.41	55.3	105.5	61.9 ± 4.77	92.3

**Table 4 molecules-27-04857-t004:** EIS parameters for mild-steel coupon without and with the addition of various concentrations of PMBMH in 1 M corrosive solution.

Conc. (M)	Rs (Ω cm^2^)	Rct (Ω cm^2^)	Cdl (μF)	IE%
Blank	2.047	54.85	493	0
0.0001	1.915	115.04	289	57.5
0.0002	1.836	158.68	311	69.6
0.0003	2.246	274.28	458	78.6
0.0004	1.703	323.8	641	87.6
0.0005	1.453	475.92	678	91.1

**Table 5 molecules-27-04857-t005:** Calculated quantum parameters of the studied inhibitor.

*E* _HOMO_	*E* _LUMO_	Δ*E* (eV)	*A*	*I*	χ (eV)	η (eV)	Δ*N* (eV)	μ (D)
−8.826	−2.336	6.49	2.336	8.826	5.581	6.49	0. 0.0586	2.7364

**Table 6 molecules-27-04857-t006:** Mulliken charges of tested inhibitor molecules.

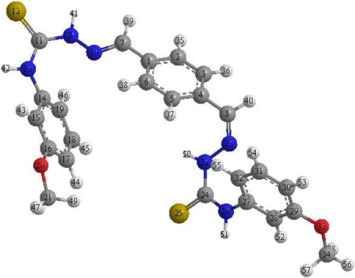									
**Atoms**	**Charges**	**Atoms**	**Charges**	**Atoms**	**Charges**	**Atoms**	**Charges**	**Atoms**	**Charges**
C(1)	−0.0273	C(8)	−0.1133	C(15)	−0.1824	N(22)	−0.2717	C(29)	0.0979
C(2)	−0.1164	N(9)	−0.2638	C(16)	0.0916	N(23)	−0.0308	C(30)	−0.1626
C(3)	−0.1069	N(10)	−0.0278	C(17)	−0.2022	C(24)	0.1863	C(31)	−0.0843
C(4)	−0.0879	C(11)	0.179	C(18)	−0.081	S(25)	−0.3009	C(32)	−0.1626
C(5)	−0.1256	S(12)	−0.2816	C(19)	−0.1548	N(26)	−0.2613	O(33)	−0.2084
C(6)	−0.0927	N(13)	−0.2595	O(20)	−0.2126	C(27)	0.0914	C(34)	−0.0773
C(7)	−0.1274	C(14)	0.0778	C(21)	−0.0781	C(28)	−0.2309	H(35)	0.1386

**Table 7 molecules-27-04857-t007:** Mild-steel chemical composition (wt%).

C	Mn	Si	Al	S	P	Fe
0.21%	0.05%	0.38%	0.01%	0.05%	0.09%	balance

## Data Availability

Not applicable.

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
