# Peer review of "Corrosion Inhibition of Mild Steel in Hydrochloric Acid Environment Using Terephthaldehyde Based on Schiff Base: Gravimetric, Thermodynamic, and Computational Studies"

_molecules, 2022, doi:10.3390/molecules27154857_

Round 1

Reviewer 1 Report

Increase the no of Keywords

delete the Table 1.

add the references below

Inhibition of corrosion: mechanisms and classifications in overview

MH Al-mashhadani, AA Ahmed, Z Hussain, SA Mohammed, RM Yusop, ... Al-Qadisiyah Journal Of Pure Science 25 (2), 1-9     INHIBITIVE EFFECT OF ATENOLOL ON THE CORROSION OF ZINC IN HYDROCHLORIC ACID AH Alwash, DH Fadhil, AA Ali, F Abdul-Hameed, E Yousif RASÄ€YAN Journal of Chemistry   Synthesis of Thiophene Derivative and its Use as Zinc Metal Corrosion Inhibitor in Hydrochloric Acid Solution N Assad, F Abdul-Hameed, A Balakit, E Yousif Journal of Al-Nahrain University 18 (2), 69

Author Response

Dear reviewer,

Thank you for useful comments. Al have been conducted. Please see the revised manuscript.

Best regards

Reviewer 2 Report

REPORTS ON: molecules-1792406

The aim of the proposed manuscript is reasonably interesting. It seems that novelty is provided and technological contribution is also attained. However, there are a great number of weaknesses that induce that the proposed manuscript NOT DESERVES its final publication.

1.                     Into the Abstract, at least three distinctive verbal tenses are used. It is suggested that only one be adopted, e.g. simple present tense.

2.                     Since the proposed title concerns to “corrosion inhibition”, at least another corrosion measurement technique should be used. At least potentiodynamic polarization and EIS (electrochemical impedance spectroscopy). This due to both techniques will more easily and it has more sensibility (accuracy) to provide analysis into a film formation. The gravimetric results could also be used as a additional information and not the main focus. A adsorption phenomenon is intrinsically and intimately associated with interfacing reactions occurring between sample and electrolyte (or medium). After this, a possible film, protective or not, and chemical or physic or chemical interactions will define the resulting inhibition. When only gravimetric analysis is used, a great number of information (mainly electrochemical and possibly chemical) is neglected.

3.                     The reproducibility into Sections 2 and 3 (Experimental procedure) is rather and poorly described. This seems that the attained results are merely speculative. No error ranges are used and demonstrated.

4.                     Why only Langmuir adsorption isotherm is used?  There are other ones, e.g. Freundlich or Flory-Huggins’ models (or other ones) that could also be compared and discussed. At least a sentence detailing and justifying their selection should be proposed/discussed.

Author Response

Dear Reviewer,

Thank you for useful comments and suggestions, all have been done point by point so please see the response letter and the revised manuscript.

Best regards

Reviewer 3 Report

The paper "Corrosion Inhibition of Mild Steel in Hydrochloric Acid Envi- 2 ronment Using Terephthaldehyde Based on Schiff Base: Gravi- 3 metric, Thermodynamic and Computational Studies" needs major improvement. One or two methods of corrosion testing of inhibitors must be added.

The paper should have, first of all, at least two testing methods of the corrosion inhibitors and their characteristics. Also, Kinetic corrosion parameters and thermodynamic parameters should be calculated with the new methods added. Surface analysis studies a (SEM / EDAX, FT-IR, e.g.) are required. In conclusion, The paper should be completely redone.

Author Response

Dear Reviewer,

Thank you for useful comments and suggestions, all have been done point by point so please see the response letter and the revised manuscript.

Round 2

Reviewer 2 Report

REPORTS ON: cmd-1845437

The proposed manuscript is reasonably well-organized and reasonably discussed. Novelty seems to be provided and a great number of experimentations are carried out. Based on the results and contribution, the manuscript seems to DESERVES its final publication after a MAJOR REVISION, as following indicated.

1.                    Firstly of all, a great number of self-citation is verified. At least 4 or 5 self-citations are not intrinsically corresponding with main text and discussion provided. It is hardly suggested that main text be meticulously revised and these “excessive” be omitted/deleted. It is hardly recognized that self-citation has no an offensive academic aspect. However, it is implicit that self-citation more than ~10% from that proposed list of reference be considered. Into the Abstract, a simple past tense should be used. At least three distinctive verbal tenses are used.

2.                    Secondly, the Abstract should be meticulously revised. Ony a simple past tense should be used. At least two verbal tense are used.

3.                    Table 3, mainly those values corresponding with “icorr” should be accompanied with error ranges.

4.                    Fig. 7 should be revised (reworked) in order to included arrows indicating each corresponding “icorr” value. Additionally, no logarithmic values are appropriated. Please, it should be revised and normal scale used. This facilitates the future readers to compare and determined correctly with Table results.

5.                     At line 452, the term “…scan rate of 0.5 mVs1”, should be revised. This mainly “mVs1” should be replaced with “mVs-1” or optionally “mV/s”. Also after this correction, the follow sentences and references should, be included after line 452:

“It is remarked that potential scan rate has an important role in order to minimize the effects of distortion in Tafel slopes and corrosion current density analyses, as previously reported [AA-CC]. However, based on these reports, the adopted 0.5 mV/s has no deleterious effects [AA,BB,CC] on Tafel extrapolations to determine the corrosion current densities of the examined samples.”

[AA] Duarte T, Meyer Y.A. Osório W.R. The Holes of Zn Phosphate and Hot Dip Galvanizing on Electrochemical Behaviors of Multicoatings on Steel Substrates. Metals 2022, 12(5): 863; https://doi.org/10.3390/met12050863

[BB] Zhang X.L., Jiang Zh.H., Yao Zh.P, Song Y., Wu Zh.D. Effects of scan rate on the potentiodynamic polarization curve obtained to determine the Tafel slopes and corrosion current density. Corrosion Science. 2009, 51: 581-587.

[CC] E. McCafferty. Validation of corrosion rates measured by Tafel extrapolation method. Corr. Scie 47 (2005) 3202-3215.

6.                    Since equivalent circuit is used and impedance parameter are show in Table 4. The follow sentence and it corresponding references should be included:

“Since an equivalent circuit is used in order to determine the simulated values and compare with experimental data, a CNLS (complex non-linear least squares) simulation is used, as previously reported [AA;DD-EE] was carried out. This should be mentioned and the follow references should be included/cited:

[DD] Metals 202212(3), 417; https://doi.org/10.3390/met12030417

[EE] Metals 202212(5), 863; https://doi.org/10.3390/met12050863

Author Response

Dear Reviewr,

Thank you for useful comments and suggestions, all have been done point by point so please see the revised manuscript.

Thank you

Best regards

Reviewer 3 Report

The  paper entitled "Corrosion Inhibition of Mild Steel in Hydrochloric Acid Environment Using Terephthaldehyde Based on Schiff Base: Gravimetric, Thermodynamic and Computational Studies" can be published in your journal.

Author Response

Dear reviewer,

Thank you for useful comment, we will send the manuscript for proofreading.

Thank you

Best regards